# Semantic Segmentation for Digital Archives of Borobudur Reliefs Based on Soft-Edge Enhanced Deep Learning

**Shenyu Ji** [1,*]**, Jiao Pan** [1]**, Liang Li** [2] ⬤**, Kyoko Hasegawa** [1]**, Hiroshi Yamaguchi** [3]**, Fadjar I. Thufail** [4]**, Brahmantara** [5]**, Upik Sarjiati** [4] **and Satoshi Tanaka** [2]

[1]  Research Organization of Science and Technology, Ritsumeikan University, Shiga 525-8577, Japan
[2]  College of Information Science and Engineering, Ritsumeikan University, Shiga 525-8577, Japan
[3]  Nara National Research Institute for Cultural Properties, Nara 630-8577, Japan
[4]  Research Center for Area Studies, National Research and Innovation Agency, Jakarta 12710, Indonesia
[5]  Borobudur Conservation Office, Magelang 56553, Indonesia
[*]  Correspondence: jishenyu@gst.ritsumei.ac.jp

**Abstract:** Segmentation and visualization of three-dimensional digital cultural heritage are important analytical tools for the intuitive understanding of content. In this paper, we propose a semantic segmentation and visualization framework that automatically classifies carved items (people, buildings, plants, etc.) in cultural heritage reliefs. We also apply our method to the bas-reliefs of Borobudur Temple, a UNESCO World Heritage Site in Indonesia. The difficulty in relief segmentation lies in the fact that the boundaries of each carved item are formed by indistinct soft edges, i.e., edges with low curvature. This unfavorable relief feature leads the conventional methods to fail to extract soft edges, whether they are three-dimensional methods classifying a three-dimensional scanned point cloud or two-dimensional methods classifying pixels in a drawn image. To solve this problem, we propose a deep-learning-based soft edge enhanced network to extract the semantic labels of each carved item from multichannel images that are projected from the three-dimensional point clouds of the reliefs. The soft edges in the reliefs can be clearly extracted using our novel opacity-based edge highlighting method. By mapping the extracted semantic labels into three-dimensional points of the relief data, the proposed method provides comprehensive three-dimensional semantic segmentation results of the Borobudur reliefs.

**Keywords:** cultural heritage; digital archive; semantic segmentation; deep learning; edge extraction from relief; Borobudur temple; Borobudur reliefs

## 1. Introduction

Recently, with the development of three-dimensional (3D) digitization of large-scale cultural heritage properties [1–4], efficient analysis of digitized properties has been the focus of an increasing amount of research [5–7]. Segmentation and visualization of three-dimensional scanned cultural heritage are important analytical tools for interpretation and intuitive understanding of the associated content. Semantic segmentation for three-dimensional point clouds has been widely investigated, such as outdoor scene understanding for autonomous driving [8–11] and robotics [12–14]. Semantic segmentation of three-dimensional digital cultural heritage has broad applications, such as the identification of architectural elements [15] or analysis of the state of conservation of materials [16]. However, automatic semantic segmentation for three-dimensional point clouds of cultural heritage is challenging because most cultural heritage properties consist of complex structures and are distinct from each other. With the development and application of deep learning in recent years, the semantic segmentation of three-dimensional point clouds in cultural heritage has made a significant breakthrough [15–17].

In this study, we focus on semantic segmentation for three-dimensional point cloud-type digital archives of cultural heritage reliefs. Cultural reliefs were often carved to

illustrate stories about historical characters or cultural events. For example, the Borobudur temple, which is a UNESCO Cultural Heritage Site in Indonesia, has 2672 bas-relief panels (sculptural reliefs in which forms extend only slightly from the background), containing 1460 narrative panels and 1212 decorative panels. Figure 1 shows a photograph of the Borobudur Temple in Indonesia. Figure 2 shows an example of the Borobudur Temple relief. These reliefs are the largest collection in the world and can be divided into five sections, each of which describes a different Buddhist story. Semantic segmentation of the reliefs can help researchers understand the figures and objects that are relevant to the stories and distinguish them from the decorative background.

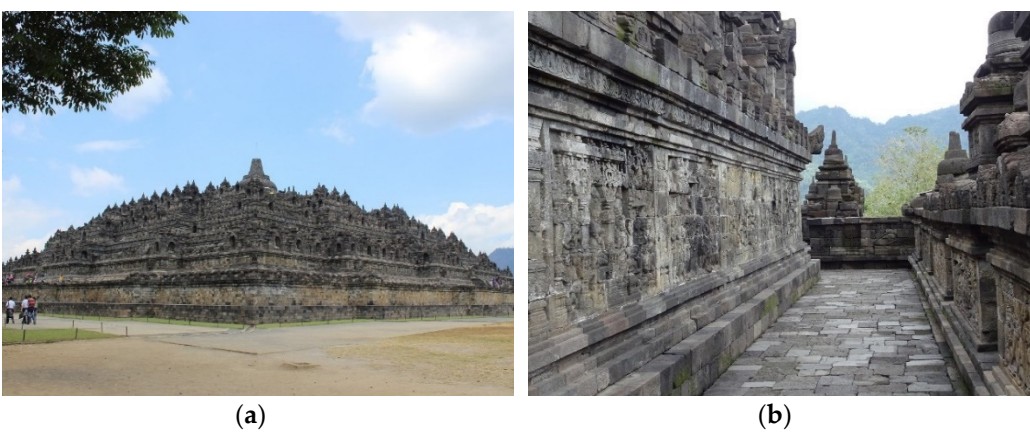

(**a**)       (**b**)

**Figure 1.** Borobudur Temple in Indonesia (photograph). (**a**) Borobudur temple; (**b**) corridors with 1460 narrative relief panels on walls. The total relief surface is 2500 square meters.

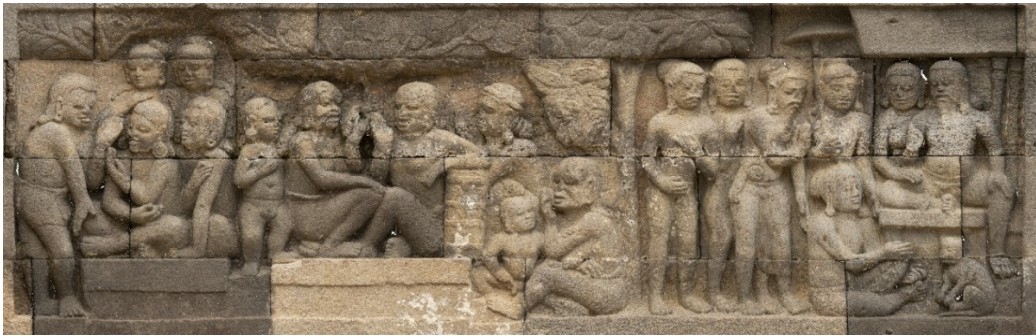

**Figure 2.** Borobudur Temple Relief, Karmawibhangga, panel No. 22, tells the story of a group of men carrying umbrellas who approached a holy man sitting in the shade of a wooden building, while people nearby were peacefully chatting. To tell these beautiful Buddhist stories, the reliefs are carved with different patterns that are raised only slightly from the background. The reliefs can be divided into several semantic labels: characters, plants, animals, ships, buildings, and so on.

However, semantic segmentation for bas-reliefs has inherent difficulties that are not present in conventional deep learning-based semantic segmentation methods, which focus on indoor or outdoor scene analysis. (1) Relief is an undulated surface different from typical three-dimensional objects that have 360-degree information. This leads to the inability of existing deep learning-based three-dimensional point cloud semantic segmentation methods to work directly on reliefs. Because supervised learning methods require a large annotated dataset, the lack of a benchmark dataset of bas-reliefs makes the task more challenging. (2) The boundaries of each carved item are formed by indistinct soft edges, i.e., edges with low curvature. This unfavorable feature of reliefs leads the conventional methods to fail to extract soft edges, whether they are three-dimensional methods classifying a three-dimensional scanned point cloud or two-dimensional (2D) methods classifying pixels in a drawn image. However, the soft edges in reliefs serve as an important cue in carved

item semantic segmentation. An effective edge highlighting method would improve the segmentation accuracy.

In this paper, we propose a semantic segmentation method that automatically classifies carved items (people, buildings, plants, etc.) in cultural heritage reliefs. This method is successfully applied to the bas-reliefs of the Borobudur Temple. To address difficulty (1), we propose a deep learning-based semantic segmentation framework for relief point clouds by projecting them into multichannel images, i.e., RGB images, depth maps, and soft-edge images. In addition, we propose an interactive relief panel extraction method to improve the accuracy of image projection, as well as the efficiency of processing large-scale point clouds during dataset preparation. For difficulty (2), we propose a method that automatically extracts the soft edges as one of the inputs in multichannel images so that the soft edges can be processed specifically. We adopt a novel three-dimensional edge detector referred to as opacity-based edge highlighting. This method is an application of the opacity-controlling mechanism employed in stochastic point-based rendering (SPBR), a high-quality transparent visualization method. The opacity-based edge highlighting method can express the soft-edge area in a relief as a gradation of surface opacity. This opacity gradation produces distinct brightness gradation in the two-dimensional image space, creating recognizable soft-edge images. We utilize the extracted soft edge as guide information in our network. An intermediate prediction of the semantic edge is arranged for the soft-edge information in the network. By supervised multitask learning, soft-edge information is effectively utilized for semantic segmentation.

## 2. Related Work

As a crucial tool regarded for analysis, segmentation and classification of digital cultural heritage have become widely investigated. Grilli et al. [16] proposed a texture-based approach and a geometry-based approach for the classification of three-dimensional data of heritage scenarios, which are effective for restoration and documentation purposes. Several works have attempted to classify cultural heritage images by employing different kinds of techniques, such as the naive Bayes nearest neighbor classifier [18], support vector machines [19], K-means algorithms [20], and latent Dirichlet allocation [21]. Deep learning has enabled a breakthrough in this work [15,22]. Pierdicca [15] improved the dynamic graph convolutional neural network (DGCNN) to segment elements, such as churches, chapels, cloisters, porticoes, and loggias in architectural cultural heritage three-dimensional point cloud data.

The three-dimensional semantic segmentation aims at classifying each point as a specific part of the point cloud. PointNet [23], an end-to-end deep neural network, is a pioneering and well known point cloud segmentation architecture that operates directly on point clouds and allows for the acquisition of input permutation invariance. PointNet++ [24] was proposed as a development from PointNet to retain local geometries and to consider local neighborhoods to ensure strong robustness and detail capture. PointNet and PointNet++ are representatives of point-based approaches [25–30]. In addition, benefiting from the convolutional neural network (CNN), projection-based [31–33], voxel-based [34,35], and multirepresentation fusion-based [36,37] methods have been proposed.

As relief is an undulated surface without 360-degree information, we consider that the semantic segmentation of the relief point cloud is suitable to be solved by the projection-based method with two-dimensional semantic segmentation. Two-dimensional semantic segmentation has been studied so successfully in computer vision [38–47] that increasing research on three-dimensional semantic segmentation is being performed by two-dimensional methods to achieve better results, e.g., projection-based methods [32]. CNNs have been widely used in two-dimensional semantic segmentation [38–45]. Existing segmentation architectures typically consist of two stages: backbone and segmentation. The first stage extracts features from RGB images, with popular models such as ResNet [38] and DenseNet [39] pretrained on the ImageNet dataset [40]. The final stage seeks to make predictions based on the extracted features. In this stage, methods such as upsampling [41],

PPM [42], and ASPP [39] are used. Note that the convolutional layers serve as the primary building blocks in both stages. CNN-based semantic segmentation in the overall framework is commonly used, such as FCN [44], SegNet [41], U-Net [45], PSPNet [42], and DeepLab [43].

We utilize a three-dimensional edge highlighting method to extract soft edges as unique features from the relief point cloud. Three-dimensional edge highlighting is more generally analyzed as the feature highlighting of three-dimensional point clouds, which are actively performed by a variety of mathematical and computational technique-based methods [48]. Recently, statistical methods using eigenvalue-based three-dimensional feature values have been widely employed [49–52]. These three-dimensional feature values are defined based on the eigenvalues of the local three-dimensional covariance matrix. We utilize an opacity-based edge highlighting method [53] to express the soft-edge area in relief as a gradation of surface opacity to confirm that the comparably higher-curvature region can also be recognized. This method is based on SPBR [54,55] to extract soft edges in relief data. SPBR is a transparent visualization method proposed in our previous work. The SPBR algorithm achieves controlled perspective visualization by interpreting opacity as the probability that each pixel in the image is the color of the input point.

## 3. Method

This section describes the framework of semantic segmentation of the Borobudur reliefs. We introduce the overview of the entire framework in Section 3.1. Our proposed framework contains four parts using different proposed methods. The detailed introductions of these methods are described from Sections 3.2–3.5.

### 3.1. Overview

We propose a single-view projection-based semantic segmentation framework for the three-dimensional point cloud of Borobudur reliefs. The entire structure of the framework is depicted in Figure 3. The proposed method projects information from a three-dimensional point cloud to multichannel images and utilizes a deep learning-based network to predict the semantic segmentation results. First, a relief panel extraction method is proposed to extract the relief, which is the component for semantic segmentation, from photogrammetric point cloud data. This relief panel extraction method, which automatically includes RGB images and depth-map projections, improves the efficiency and accuracy of dataset creation for semantic segmentation. Then, the opacity-based edge highlighting method is utilized to extract the soft-edge information, which is the unique feature of the Borobudur reliefs from the point cloud. After the RGB images, depth maps, and soft-edge images are prepared, we implement a semantic segmentation network based on CNN deep learning named the soft-edge enhanced network. The prepared three kinds of images are concatenated into multichannel images and used as the input of the proposed network. The use of multichannel images improves the accuracy of the proposed semantic segmentation network from different perspectives. First, RGB images are used to project color information from point clouds and convert photogrammetric point clouds into photographs. Second, the depth map addresses the loss of depth information due to the projection. Third, soft-edge images can reflect the slight change in curvature in the relief point clouds. With the proposed method, the corresponding semantic labels of each carved item in reliefs can be extracted. Finally, by mapping the extracted semantic labels into three-dimensional points of the relief data, the proposed method provides comprehensive three-dimensional semantic segmentation results for the Borobudur reliefs.

### 3.2. Relief Panel Extraction and RGB-D Image Projection

The three-dimensional point cloud of the Borobudur reliefs is obtained by photogrammetry. In the raw data of the scanned points, regions other than relief panels are also included. Therefore, a relief panel extraction method is proposed in this paper. Open-source software, such as CloudCompare [56] and Meshlab [57], can be used to manually

extract relief panels. However, manual extraction and adjustment require considerable labor, and the accuracy of image projection cannot be guaranteed. In this work, we propose a semiautomatic method for relief panel extraction and RGB-D image projection. Figure 4 shows the process of relief panel extraction.

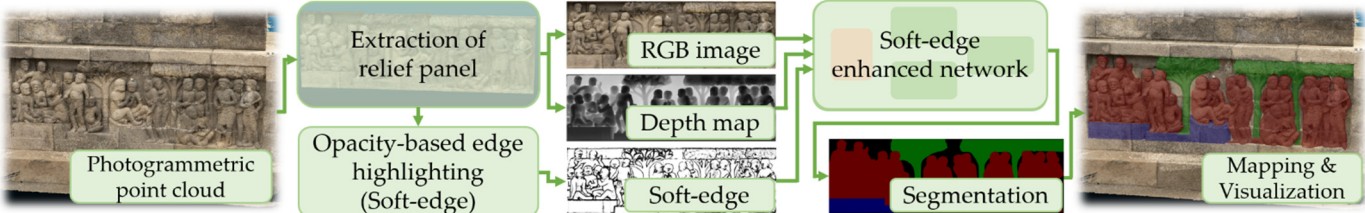

**Figure 3.** Block diagram: The first step is to extract the relief panel from the input and project the RGB image and depth map. Then, the soft edge is extracted by opacity-based edge highlighting. These images are transmitted to the soft-edge enhanced network as a multichannel to predict the two-dimensional semantic segmentation labels. Finally, the results are obtained by mapping the extracted semantic labels onto three-dimensional points of the relief data.

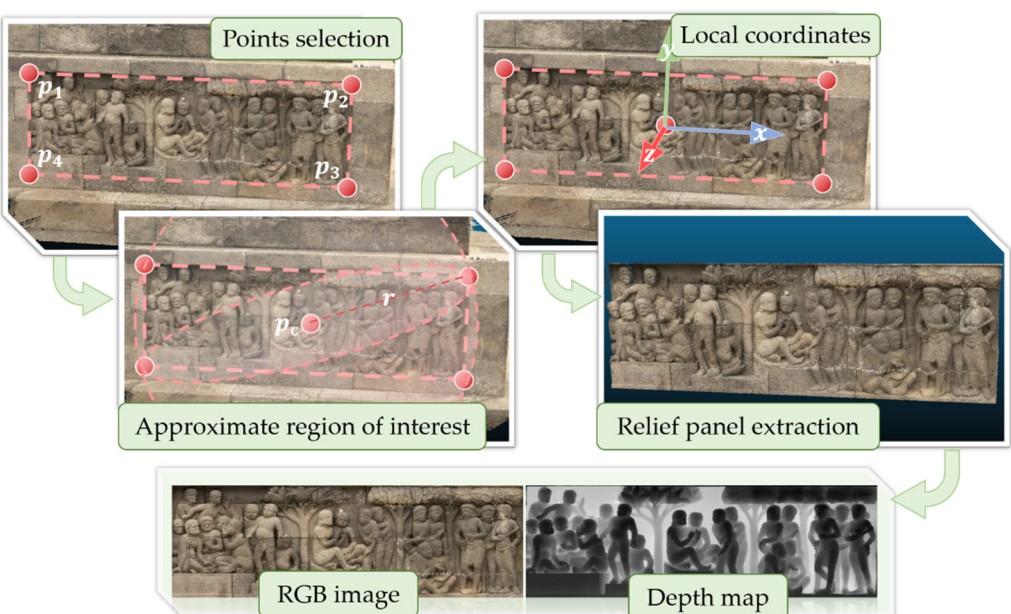

**Figure 4.** Relief panel extraction.

Control point selection. The position, size, and direction of the targeted relief panel are determined by selecting four control points. The four points are the four corners of the relief panel and are interactively selected by the user in the clockwise direction starting from the top left corner. Here, the selected points are represented as $p_1$, $p_2$, $p_3$, and $p_4$.

Approximate region of interest extraction. This process removes surrounding points and extracts an approximate region of interest around the target relief panel from the raw point cloud data. The average coordinates, $p_c$, of the four control points are used as the center of the spherical region of interest, while the farthest distance from the centroid among the four control points is utilized as the radius $r_s$ of the spherical region. Here, $p_c$ and $r_s$ are calculated using the following equations:

$$p_c = \frac{p_1 + p_2 + p_3 + p_4}{4} \tag{1}$$

$$r_s = \text{MAX}(\|p_i - p_c\|), \ p_i \in \{p_1, p_2, p_3, p_4\}. \tag{2}$$

Determination of the relief panel's local coordinates. Calculating the local coordinates of the relief panel is necessary for correct image projection. This step focuses on calculating the normal vector $n$ of the region of interest by plane fitting. After the second step, most undesired points are trimmed, leaving only the data containing the entire point cloud of the relief panel and some surrounding parts. The normal of the fitting plane is used to set the z-axis of the relief panel's local coordinate system. The positive direction of the normal is determined by the left-hand rule based on clockwise rotation of the four control points. The vertical direction from the center point $p_c$ to the line between the first and second control points is set to be the y-axis of the local coordinate system. The orthogonal x-axis is then set accordingly. After the local coordinate system is determined, the global coordinates of the point cloud can be transformed to local coordinates using the coordinate transformation

$$C_{\textbf{local}} = \mathbf{R}\,\mathbf{T}\,C_{\textbf{global}} \tag{3}$$

where $C_{\textbf{global}}$ are the global coordinates of the points in the relief panel's point cloud, and $C_{\textbf{local}}$ are the local coordinates. $\mathbf{T}$ is the translation matrix, which translates the origin of the local coordinate system to the origin of the global coordinate system. $\mathbf{R}$ is the rotation matrix, which transforms the direction between the two coordinate systems. We utilize a gradually decreasing rectangle to the region of interest to update the local coordinate system automatically.

Relief panel extraction and image projection. After the optimized local coordinate system is determined, the projection of the four control points on the x-y plane forms a quadrilateral. This quadrilateral defines the final region of interest of the target relief panel. Finally, the extracted relief panel is projected to a RGB image and a depth map. The intensity in the depth map is calculated from the value of the z-axis by a linear transformation. The intensity is set in the range of 0 to 255.

### 3.3. Opacity-Based Edge Highlighting

This section describes our proposed soft-edge extraction method for Borobudur reliefs. Figure 5 shows the process of soft-edge extraction from relief photogrammetric point cloud. We proposed this opacity-based edge highlighting method in our previous work [54]. The main idea of this method is that we extract the three-dimensional edge areas and then execute rendering with higher opacity to the three-dimensional edges to make them brighter than the surrounding non-edge regions.

The opacity is based on SPBR [54]. In this rendering method, the transparency of a visualized surface is achieved based on probabilistic theory. That is, we define the surface opacity by the probability that each pixel becomes the point color [55]. SPBR realizes fast and precise three-dimensional see-through imaging transparent visualization. As the transparency originates from the stochastic determination of pixel intensities, it achieves the correct depth feel in visualization without requiring time-consuming depth sorting of three-dimensional points. Thus, this method is particularly suitable for large-scale three-dimensional point clouds.

The steps to execute SPBR are as follows:

STEP 1. Creation of point ensembles: the point cloud is randomly divided into multiple subgroups, each of which has the same number of three-dimensional points. We call the subgroups "point ensembles." The number of ensembles is denoted as $L$.

STEP 2. Point projection per point ensemble: for each point ensemble in step 2, its constituent three-dimensional points are independently projected into the image plane. As a result, $L$ intermediate images are created. In the projection process, we incorporate the point–occlusion effect per pixel.

STEP 3. Averaging the intermediate images: an average image is created by averaging the intensities of the corresponding pixels in the above $L$ intermediate images. Then, the transparent image of the point ensembles can be created. The ensemble number $L$ controls the statistical accuracy and works as an image-quality parameter.

Considering a local area with an area of $S$, we execute the point–density adjustment for this area, such that the following user-defined surface opacity $\alpha$ is realized:

$$\alpha = 1 - (1 - s/S)^{n_{adj}/L} \tag{4}$$

where $s$ is the particle cross-section area, which is tuned such that an image of each three-dimensional point overlaps exactly one pixel; $n_{adj}$ is the adjusted number of three-dimensional points in this local area; and $L$ is the number of point ensembles.

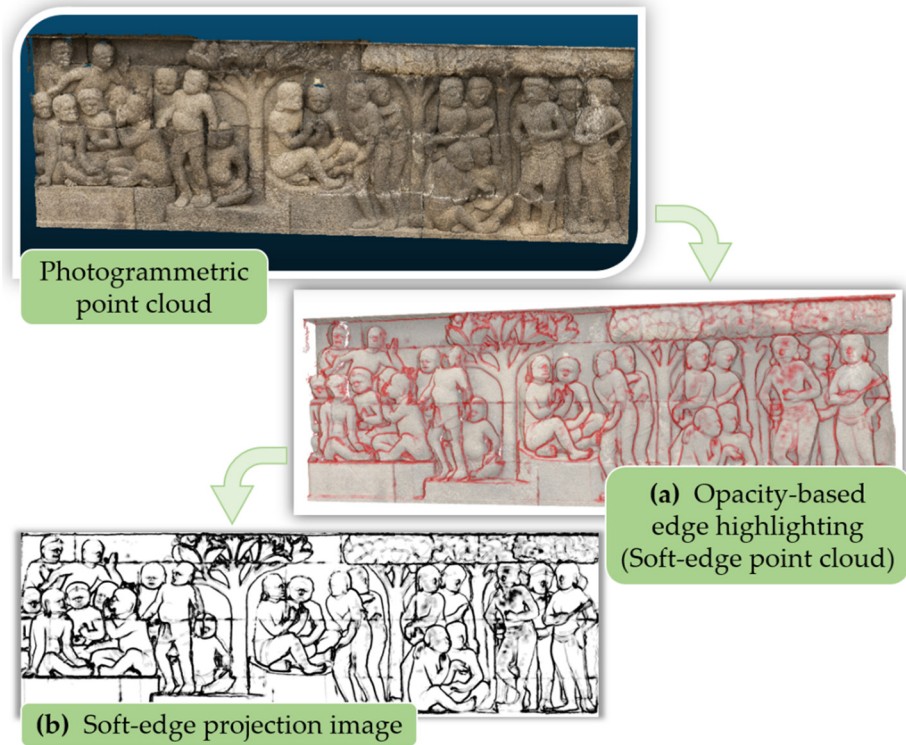

**Figure 5.** Soft-edge extraction from relief photogrammetric point cloud by opacity-based edge highlighting and image projection.

Based on this opacity provided by SPBR, we can execute our proposed opacity-based edge highlighting method. To extract the three-dimensional edge, the eigenvalue-based three-dimensional feature is utilized to extract high-curvature areas from the given point cloud. Several kinds of eigenvalue-based three-dimensional feature values have been proposed. In our research, we adopt the change in curvature and the linearity:

$$\text{Change of curvature}: \quad C_\lambda = \frac{\lambda_3}{\lambda_1 + \lambda_1 + \lambda_3} \tag{5}$$

$$\text{Linearity}: \quad L_\lambda = \frac{\lambda_1 - \lambda_2}{\lambda_1} \tag{6}$$

where $\lambda_1$, $\lambda_2$, and $\lambda_3$ are the eigenvalues of the three-dimensional structure tensor (the local three-dimensional covariance matrix) [58] with $\lambda_1 \geqslant \lambda_2 \geqslant \lambda_3 \geqslant 0$. In our implementation, the three-dimensional structure tensor is calculated for each spherical local region centered at each point. The change in curvature $C_\lambda$ measures the minimal extension of the local point distribution that should vanish in the case of a planar distribution. The linearity, $L_\lambda$, measures the difference between the two independent-directional largest extensions of the local point distribution that should also vanish in the case of a planar distribution. Thus, a large value of $C_\lambda$ or $L_\lambda$ indicates that the examined local set of points forms a three-dimensional edge of the point-based surface.

We denote the adopted three-dimensional feature value as $f$ and normalize it to make the distribution fall between zero and one. With the three-dimensional feature value $f$, we propose a function to relate a three-dimensional feature value to opacity $\alpha$. The function can highlight the soft edge from the relief point cloud. In this function, we define the opacity $\alpha$ as follows:

$$\alpha(f) = \begin{cases} \alpha_{\min} & (f \leqslant f_{\text{th}}) \\ \frac{\alpha_{\max} - \alpha_{\min}}{(F_{\text{th}} - f_{\text{th}})^d}(f - f_{\text{th}})^d + \alpha_{\min} & (f_{\text{th}} < f \leqslant F_{\text{th}}) \\ \alpha_{\max} & (F_{\text{th}} < f \leqslant 1) \end{cases} \tag{7}$$

where $f_{th}$ and $F_{th}$ are two threshold values with $f_{\text{th}} \leqslant F_{\text{th}}$. The parameter $d$ controls the speed of the opacity increase. The opacity graph is shown in Figure 6. In the soft-edge regions, we need a plateau area at $f > F_{\text{th}}$ to cover all of the highest curvature regions. Thus, we assign a constant high opacity for local areas with $f > F_{\text{th}}$. To make the opacity gradually decrease as the position moves away from the centerline of the edge regions, we make $\alpha$ gradually increase for local areas with $f_{\text{th}} < f \leqslant F_{\text{th}}$. This opacity distribution gives the narrow areas around the centerlines comparably higher opacity, and the areas are highlighted with brighter colors.

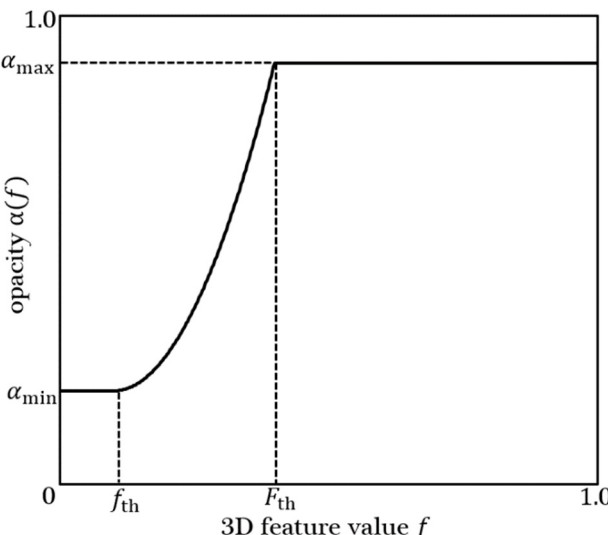

**Figure 6.** Typical relation between the three-dimensional feature value $f$ and opacity $\alpha$ when adopting the function.

The visualization of soft-edge highlighting from relief photogrammetry by the opacity-based method is shown in Figure 5a. The soft edge is a unique feature in the reliefs that forms boundaries between the carved items and the wall background. Soft edges have only slightly higher curvatures than their surrounding areas. This method successfully extracts the soft-edge features in the relief.

After the soft-edge component is extracted by the proposed method, we project it into two-dimensional images as the input for the network. The value of each pixel in the projected image is converted from the opacity of the extracted three-dimensional edge points, as shown in Equation (8).

$$g(\alpha) = 1 - \left( \frac{\alpha - \alpha_{\min}}{\alpha_{\max} - \alpha_{\min}} \right) \tag{8}$$

where $g$ represents the value of each pixel in the projected soft-edge image, and $\alpha$ represents the opacity of the extracted soft-edge points. The soft-edge projection image is shown in Figure 5b.

### 3.4. Soft-Edge Enhanced Network for Semantic Segmentation

In this section, we present our proposed soft-edge enhanced semantic segmentation network. The proposed method is based on an encoding–decoding structure, as shown in Figure 7. Section 3.4.1 presents the soft-edge enhanced utilization techniques of the semantic segmentation network. The structure of our soft-edge enhanced network is described in Section 3.4.2. The training loss function is illustrated in Section 3.4.3.

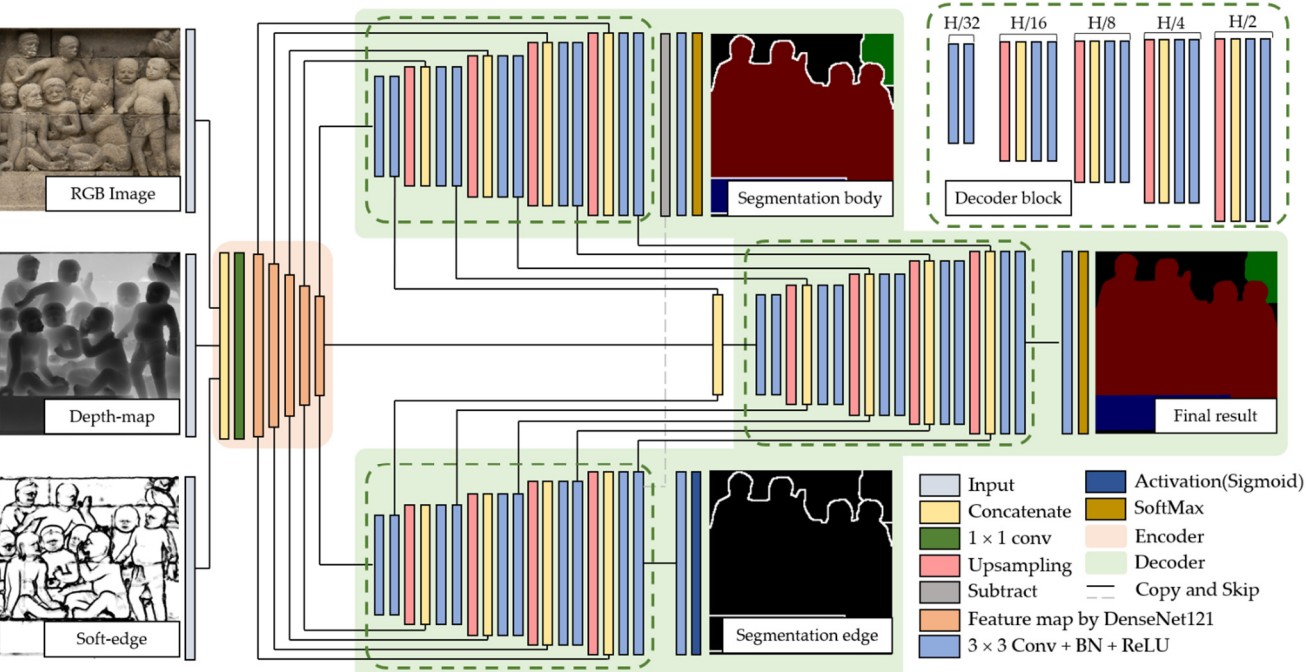

**Figure 7.** Structure of the soft-edge enhanced network. This is a multichannel input network with RGB images, depth maps, and soft-edge images. The network utilizes DenseNet121 as the backbone and extracts five sizes of feature layers: H/2, H/4, H/8, H/16, and H/32. Three decoder blocks (as shown in the upper right) are set to supervise the segmentation edge and segmentation body intermediate layer and semantic segmentation results in the decoder sections.

### 3.4.1. Soft-Edge Enhanced Utilization

As the soft edge is a unique feature of relief, we utilize two techniques to effectively apply the soft-edge information. (1) Feature skip connection. This technique is inspired by U-Net [45], a pioneering method in biomedical image segmentation, which is similar to relief analysis, with fewer training data and focusing on high-frequency information, such as edges. During deep convolution, the network tends to focus more on wide perception and ignore the edge detail information. The feature skip connection can recover and retrace edge detail features to enhance the soft-edge information. (2) Multitask learning to supervise the intermediate layer. This technique is inspired by DecoupleSegNets [59], which is based on the concepts of semantic bodies and edges. We utilize a similar idea that decouples the body and the edge with different types of supervision by multitasking to optimize the edge feature and the body feature. The edge feature is supervised by the segmentation edge mask, while the body feature is supervised by the mask where the segmentation edges are ignored during training. Compared with DecoupleSegNets, we focus on edge prediction, while DecoupleSegNets is oriented more to body prediction. Our use of soft-edge information extracted from the three-dimensional edge highlighting as input can facilitate network prediction of the segmentation edges from soft-edge information. Supervising the intermediate layer of the segmentation edge creates an easier task for soft edge information to maintain the clarity of the edge for segmentation and the consistency of body parts for each object.

3.4.2. Network Structure

The network is a multichannel input network with RGB images, depth maps, and soft-edge images. The input layers of the network consist of three different images with a resolution of $H$, as shown in Figure 7. The RGB image input layer has three channels, and the depth-map input layer and soft-edge input layer have one channel.

At the beginning of the encoder phase, these three input images are concatenated into one integrated feature map with five channels. Then, a subsequent $1 \times 1$ convolutional layer is utilized to output a feature map that self-adaptively adjusts the number of channels to three channels. The following parts of the encoder contain five layers that are extracted from DenseNet121 [39], with $H/2$, $H/4$, $H/8$, $H/16$, and $H/32$. The channel numbers of each layer are 64, 64, 128, 256, and 512.

Three decoder stages that have similar decoder blocks are utilized for the semantic body, edge, and final output. The decoder block first passes the input features through the $3 \times 3$ convolutional layer twice with a size of $H/32$. This is followed by four upsampling steps. After each upsampling layer, the feature layer is concatenated with the incoming feature layer of the same size by skip connection, and the features are passed through the $3 \times 3$ convolution layer twice. The output of the decoder block is the same size as $H/2$ with 64 channels.

The segmentation edge decoder passes the features of the output of the decoder block into a $3 \times 3$ convolution layer that outputs with the size $H/2$ with 1 channel. The sigmoid activation layer is applied to predict the probability of the pixels being inside the segmentation edge region. The segmentation edge consists of the nonzero pixels obtained from performing high-pass filtering on the semantic segmentation result.

In the segmentation body decoder, the output features of the decoder block are decoupled by subtracting the output of the segmentation edge decoder block. A $3 \times 3$ convolution layer follows the decoupled feature layer that outputs with the size $H/2$ with $N_{cls}$ channels, where $N_{cls}$ represents the number of classes. Then, a softmax layer is used to determine the probability of each pixel belonging to each class in the segmentation body. The segmentation body consists of the pixels where the segmentation edges are ignored. The learning objective $F_{body}$ for the segmentation body decoder is given by:

$$F_{body} = F_{gt} - F_{edge} \tag{9}$$

where $F_{gt}$ represents the ground truth of the semantic segmentation, and $F_{edge}$ represents the segmentation edge.

In the final decoder, every two intermediate feature maps extracted by $3 \times 3$ convolution layers in the segmentation edge and body encoder are involved in the convolution in this decoder block by a skip connection. There is also a $3 \times 3$ convolution layer and a SoftMax layer following. The $3 \times 3$ convolution layer outputs the size $H/2$ with $N_{cls}$ channels, and the softmax layer determines the probability of each pixel belonging to each class in the semantic segmentation result.

3.4.3. Training Loss

To improve the final segmentation accuracy, three loss functions are utilized for the segmentation edge decoder, segmentation body decoder, and final decoder in our paper. These three loss functions are presented as $L_{edge}$, $L_{body}$, and $L_{final}$. Then, a joint loss function L is defined in Equation (10) to train the proposed network. The total training loss $L$ is defined as follows:

$$L = \alpha L_{edge}\left(s_{edge}, \, s'_{edge}\right) + \beta L_{body}\left(s_{body}, \, s'_{body}\right) + \gamma L_{final}\left(s, \, s'\right) \tag{10}$$

where $s'_{edge}$, $s'_{body}$, and $s'$ represent the ground truth of the segmentation edge, the segmentation body, and the final result, respectively; $s_{edge}$, $s_{body}$, and $s$ represent the intermediate prediction results from the semantic edge, the segmentation body, and the final decoders,

respectively; and $\alpha$, $\beta$, and $\gamma$ are three hyperparameters that control the weighting among the three losses. We set $\alpha = 1.5$, $\beta = 1.0$, and $\gamma = 1.0$ as default.

In the segmentation edge decoder, the MSE loss is utilized to predict the probability that a pixel point is inside the segmentation edge region. The expression of $L_{\text{edge}}$ is as follows:

$$L_{\text{edge}}\left(s_{\text{edge}},\ s'_{\text{edge}}\right) = \left(s_{\text{edge}} - s'_{\text{edge}}\right)^2 \tag{11}$$

We utilize the focal loss [60] for the probability of segmentation map prediction in the segmentation body and the final decoders. The $L_{\text{body}}$ can be defined by Equation (12):

$$L_{\text{body}}\left(s_{\text{body}},\ s'_{\text{body}}\right) = \text{FL}\left(s_{\text{body}} - M_{\text{edge}},\ s'_{\text{body}} - M_{\text{edge}}\right) \tag{12}$$

$$L_{\text{final}}\left(s,\ s'\right) = \text{FL}\left(s,\ s'\right) \tag{13}$$

where $M_{\text{edge}}$ represents the ground truth of the segmentation edge as a mask. The focal loss is an extension of the cross-entropy loss function that downweights easy examples and focuses training on hard negatives.

$$\text{FL}(p_t) = -\alpha(1 - p_t)^\gamma \log(p_t) \tag{14}$$

where $p_t$ represents the probability for each class, and where $\alpha$ and $\gamma$ are two hyperparameters; $\alpha$ is a weighting factor that controls the balance among each class, and $\gamma$ smoothly adjusts the rate at which easy examples are downweighted.

### 3.5. Mapping and Visualization

We achieve the three-dimensional semantic segmentation results by mapping the two-dimensional predicted results of the network into the three-dimensional point cloud of the relief panels that are extracted from Section 3.2. The two-dimensional results overlap with the plane of $z = 0$ in the local coordinate system of the relief. The three-dimensional segmentation results in global coordinates in the photogrammetric point cloud data are obtained from the following equation.

$$C_{\textbf{global}} = \mathbf{T}^{-1}\,\mathbf{R}^{-1}\,C_{\textbf{local}}, \tag{15}$$

where $C_{\textbf{global}}$, $C_{\textbf{local}}$, $\mathbf{T}$, and $\mathbf{R}$ appear in Equation (3) in Section 3.2.

The alpha blending method is utilized to visualize the three-dimensional semantic segmentation result following Equation (16).

$$c_{\text{vis}} = \alpha c_{\text{cls}} + (1 - \alpha)c_{\text{rgb}}, \tag{16}$$

where $\alpha$ represents an opacity from 0 to 1 and where. $c_{\text{vis}}$, $c_{\text{cls}}$, and $c_{\text{rgb}}$ represent the colors of the visualization, segmentation, and original point, respectively.

## 4. Experimental Results

In this section, we introduce the process of building the dataset for semantic segmentation in Section 4.1. Section 4.2 describes our experimental approach and experimental results for the proposed network. The visualization results of the semantic segmentation for the Borobudur reliefs are shown in Section 4.3.

### 4.1. Dataset Creation

The training dataset is composed of pairs of RGB images, depth maps, soft-edge images, and segmentation labels. The dataset creation process is shown in Figure 8. The original data are photogrammetric point cloud data that contain the reliefs and structures of the temple. We collect the relief parts that contain three-dimensional coordinates and color information and separate the information into an RGB image and the corresponding depth map by the method introduced in Section 3.2. Moreover, we apply the edge highlighting

method introduced in Section 3.3 to extract the soft-edge points and project them into images. We draw the semantic segmentation category labels on the two-dimensional image and set the category to four classes: background (black), characters (red), plants (greed), and others (blue). We annotate 26 images by manually drawing object contours and giving labels to each object.

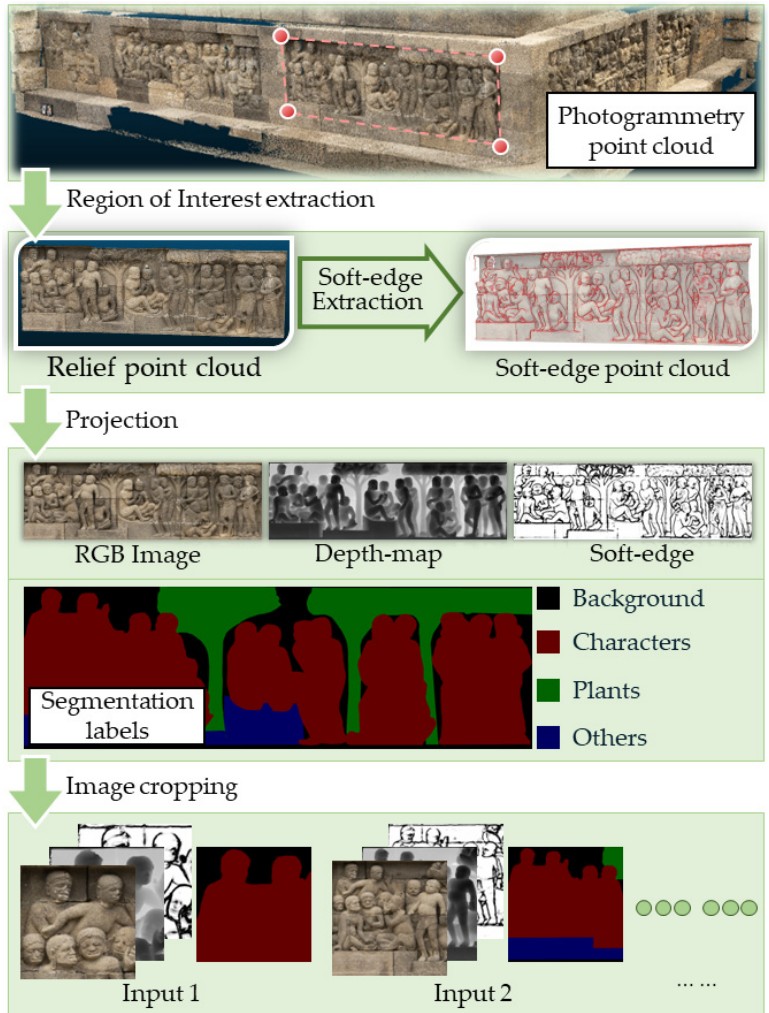

**Figure 8.** Block diagram of dataset creation.

The resolution of the projected images is 3200 × 1024, which incurs a considerable computational cost to the network. There are only 26 pairs of these projected images, which are not enough to train a deep neural network. Therefore, we cut these images into 6656 patch pairs to train the neural network. The resolution of the image patch is set by three different sizes: 1024 × 1024, 768 × 768, and 512 × 512. Among 26 pairs of projected images, 24 pairs are applied to train the neural network, and two pairs serve as validation data for the quantitative evaluation.

### 4.2. Network Evaluation

To evaluate the proposed method, two relief panels were chosen for the quantitative and qualitative experiments. From left to right in Figure 9 are the RGB image, depth map, soft edge, ground truth of semantic segmentation, and prediction results. The soft edges feature a difference in concentration. The extraction of soft edges is currently affected by the point cloud density, requiring manual fine-tuning of parameters. However, the experimental results show that our network is robust for different concentrations in soft

edges and maintains the clarity of the segmentation edge and the consistency of body parts for each object.

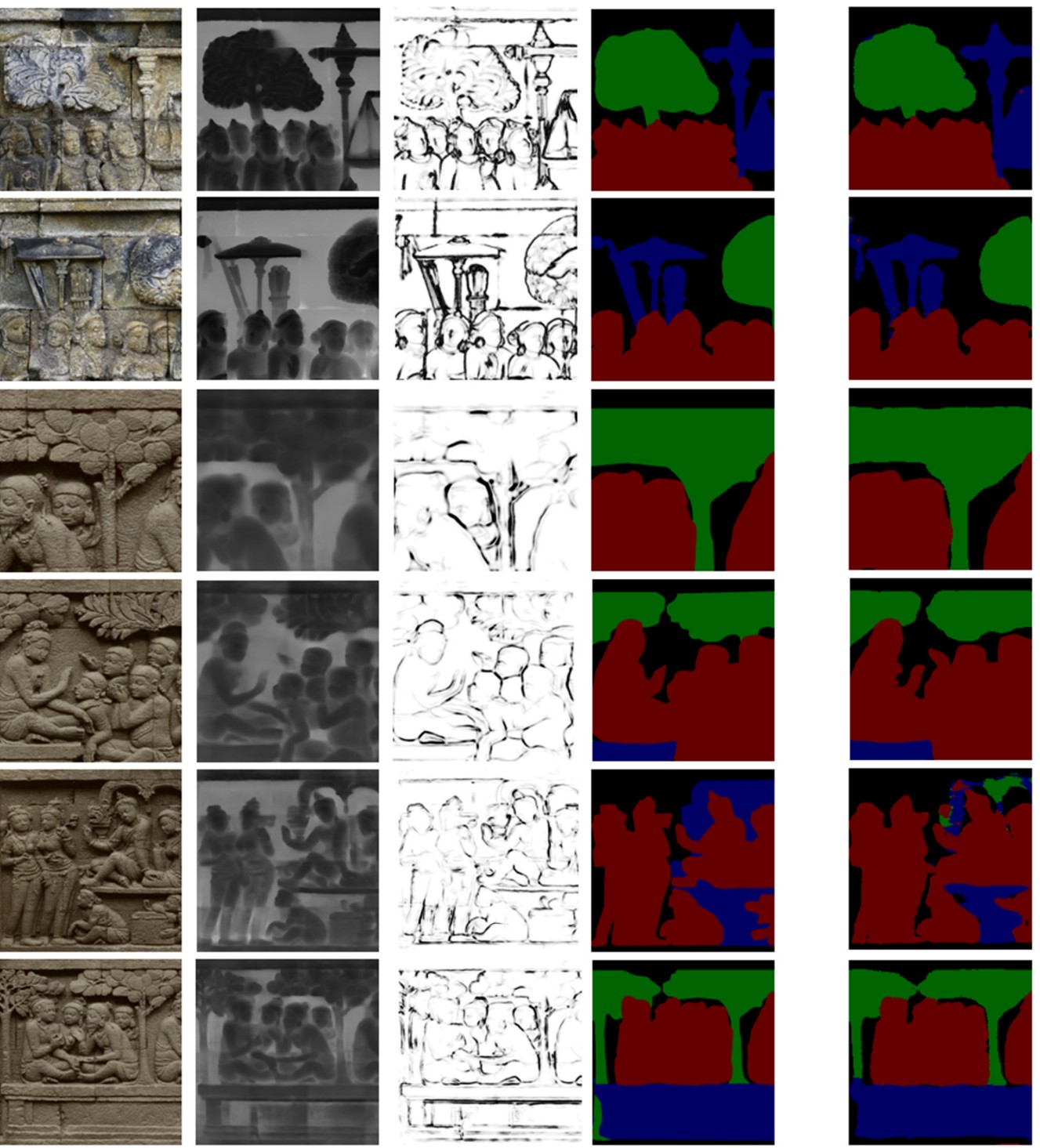

**Figure 9.** Predicted results on test data. Left to right: input RGB image, depth map, soft-edge, ground truth, and prediction results. The results show that our network is robust to different concentrations in soft edges.

We employed five metrics (accuracy, precision, recall, intersection over union (IoU), and F1-score) and output the confusion matrix to evaluate our network. The results are presented in Table 1. The left side of the table is the confusion matrix, where each row represents the pixel count of each category in the ground truth, and each column represents

the pixel count of each category in the predicted results. The right side of the table shows the evaluation metrics on each category; higher values are better. According to the evaluation metrics, our network achieves 90.53% accuracy in prediction, as well as a 81.52% mean IoU.

**Table 1.** Confusion matrix and evaluation metrics of the network on test data.

| Matrix | Background | Characters | Plants | Others | Recall | Precision | IoU | F1-Score |
|---|---|---|---|---|---|---|---|---|
| Background | 6,036,161 | 100,867 | 70,766 | 443,841 | 0.9075 | 0.8983 | 0.8230 | 0.9029 |
| Characters | 129,260 | 7,635,226 | 19,573 | 114,388 | 0.9667 | 0.9376 | 0.9083 | 0.9519 |
| Plants | 144,168 | 63,712 | 2,689,938 | 34,097 | 0.9175 | 0.8931 | 0.8267 | 0.9051 |
| Others | 409,643 | 343,360 | 231,605 | 3,771,091 | 0.7930 | 0.8643 | 0.7051 | 0.8271 |
| Mean | Accuracy: 0.9053 | | | | 0.8961 | 0.8983 | 0.8158 | 0.8968 |

We also implemented prediction for the segmentation edge information and output the image of the intermediate layer of the segmented edge. The results are shown in Figure 10. Regarding the prediction of edge information, the first column of the figure shows the segmentation edge obtained from the semantic segmentation ground truth, and the second column shows the probability of segmentation edge pixel prediction. The third column shows the ground truth, and the fourth column shows the prediction results of semantic segmentation. We consider that learning the segmentation edges helps the network clarify the contours of semantic segmentation.

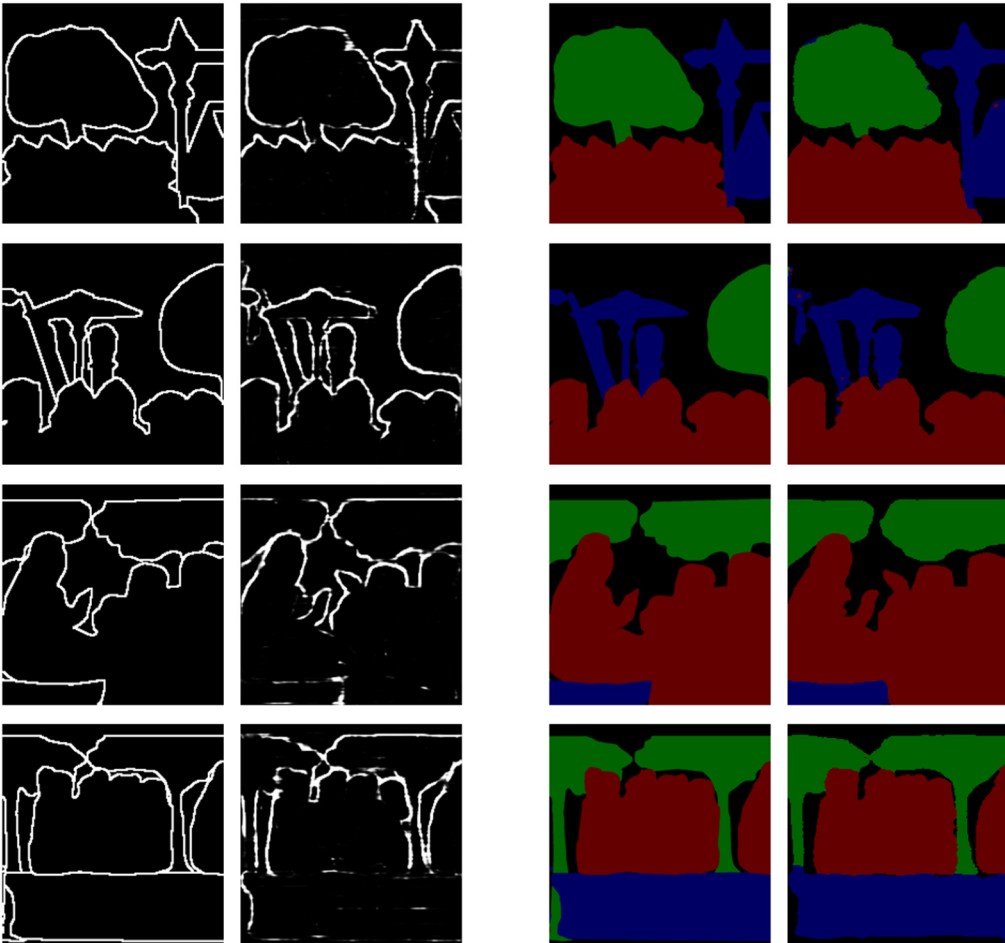

**Figure 10.** Prediction for segmentation edges. Left to right: ground truth of the segmentation edge, prediction results of the segmentation edge, ground truth of semantic segmentation, and prediction results of semantic segmentation. The results show that the network can predict approximate segmentation edges and is consistent with the prediction results of semantic segmentation.

We compare our proposed network with the existing two-dimensional semantic segmentation networks SegNet, U-Net, PSPNet, and DeepLabV3+.

The results in Figure 11 show that our proposed network prediction maintains the clarity of the segmentation edge and the consistency of semantic segmentation for each object. A comparison of the evaluation metric results is shown in Table 2.

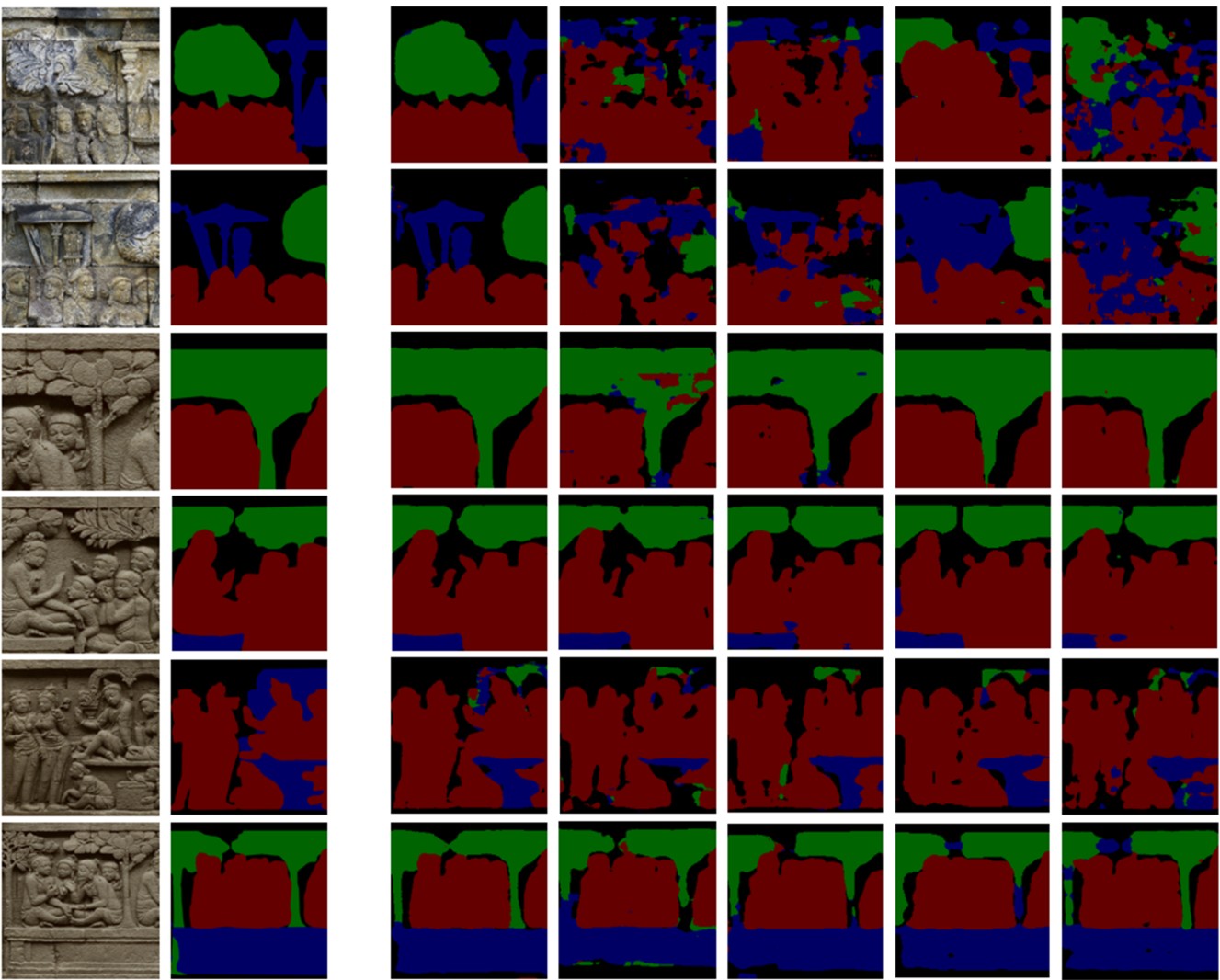

**Figure 11.** Left to right: input RGB image, ground truth of semantic segmentation, prediction results of our proposed network, and prediction results of SegNet, U-Net, PSPNet, and DeepLabV3+. The results show that extant networks are more sensitive to the color information of input images and demonstrate the robustness of our proposed network to photo information.

**Table 2.** Comparison of evaluation metric results between current networks and our proposed network. The results demonstrate that our network outperforms other networks in all metrics.

| Network | Recall | Precision | mIoU | F1-Score | Accuracy |
|---------|--------|-----------|------|----------|----------|
| SegNet | 0.6843 | 0.7079 | 0.5388 | 0.6932 | 0.7240 |
| U-Net | 0.6708 | 0.7084 | 0.5251 | 0.6839 | 0.7120 |
| PSPNet | 0.6953 | 0.7096 | 0.5470 | 0.7011 | 0.7184 |
| DeeplabV3+ | 0.6707 | 0.6877 | 0.5200 | 0.6777 | 0.6993 |
| Ours | 0.8961 | 0.8983 | 0.8158 | 0.8968 | 0.9053 |

### 4.3. Visualization Results

We used a voting algorithm for the divided patches to predict the semantic segmentation results for the whole panel of reliefs. The method mentioned in Section 3.5 used to map the visualization results on the region of interest relativity coordinate.

We performed prediction experiments for semantic segmentation of the whole relief's photogrammetric point cloud data. The point cloud of the relief is shown at the top of Figure 12. We transform the point cloud into a 3200 × 1024 image using the method described in Section 3. After the point cloud is imaged, we crop the image into *N* equally spaced and overlapping square 1024 × 1024 image patches. *N* is usually set to 64. The semantic segmentation results of these *N* patches are then predicted. Voting is performed on the overlapping parts. The two-dimensional prediction results are shown in the middle part of Figure 12. The final mapping result is shown at the bottom of the figure. Figure 13 shows the visualization results of some additional samples of relief point clouds. Our results achieve very clear classification and high accuracy and improve the readability of the relief content. We also conducted experiments to apply our proposed semantic segmentation framework directly to the original point cloud. The visualization results are shown in Figure 14.

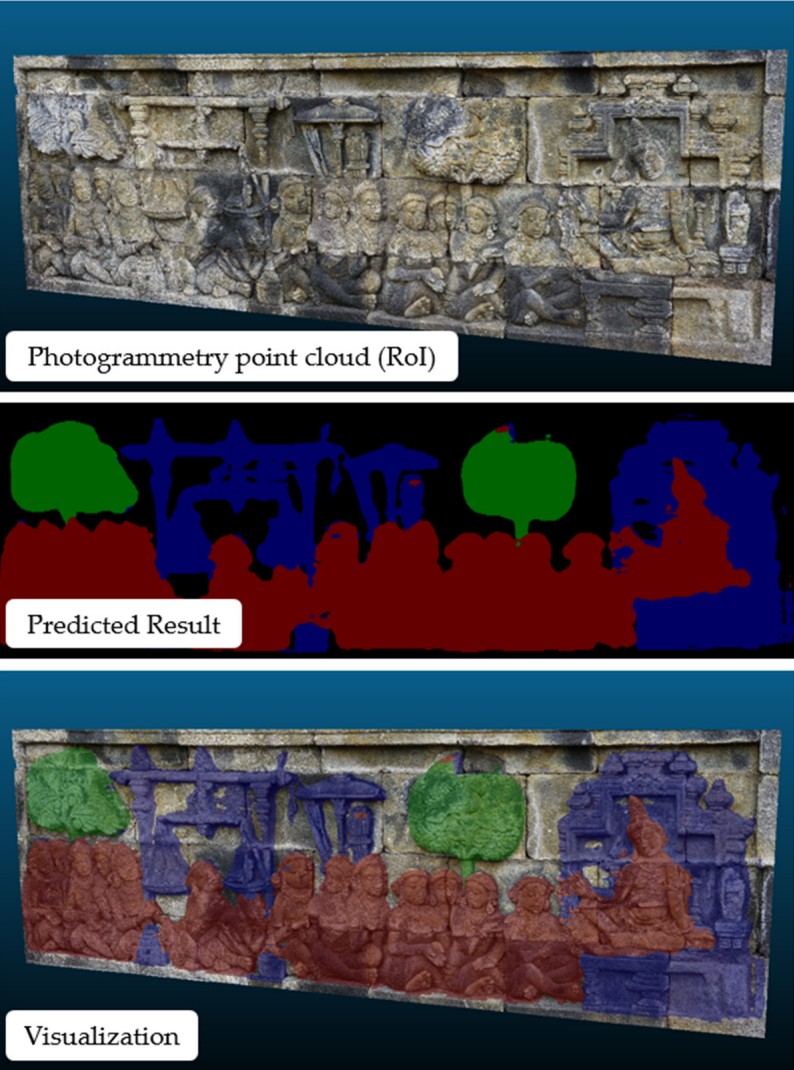

**Figure 12.** Visualization results of mapping two-dimensional prediction results to three-dimensional point cloud.

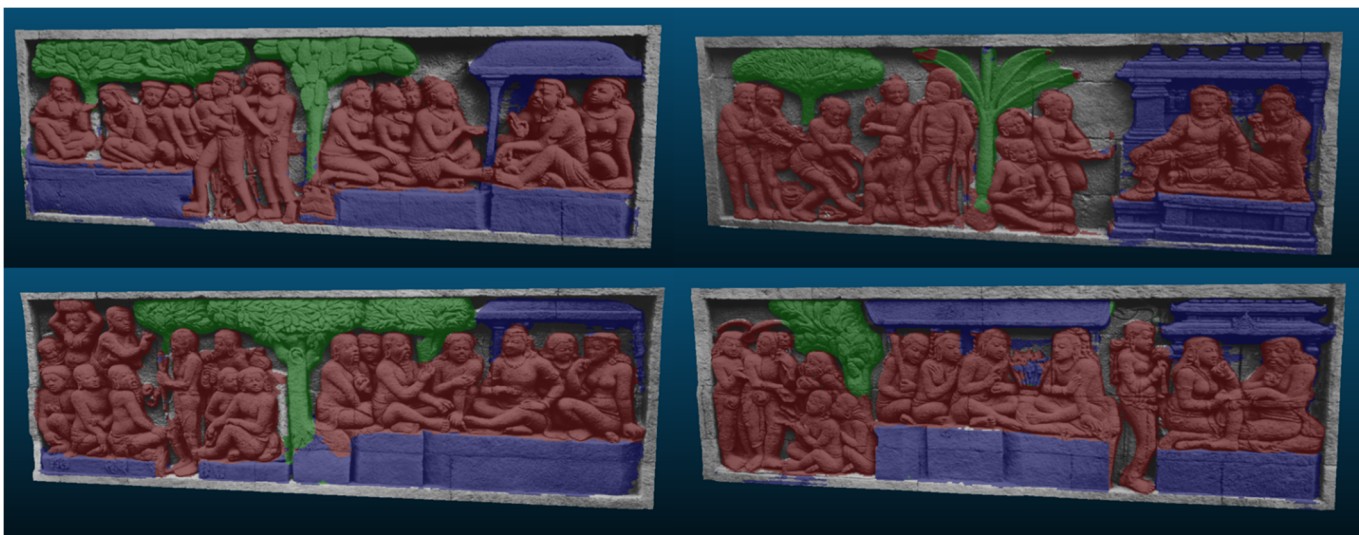

**Figure 13.** Samples of visualization results.

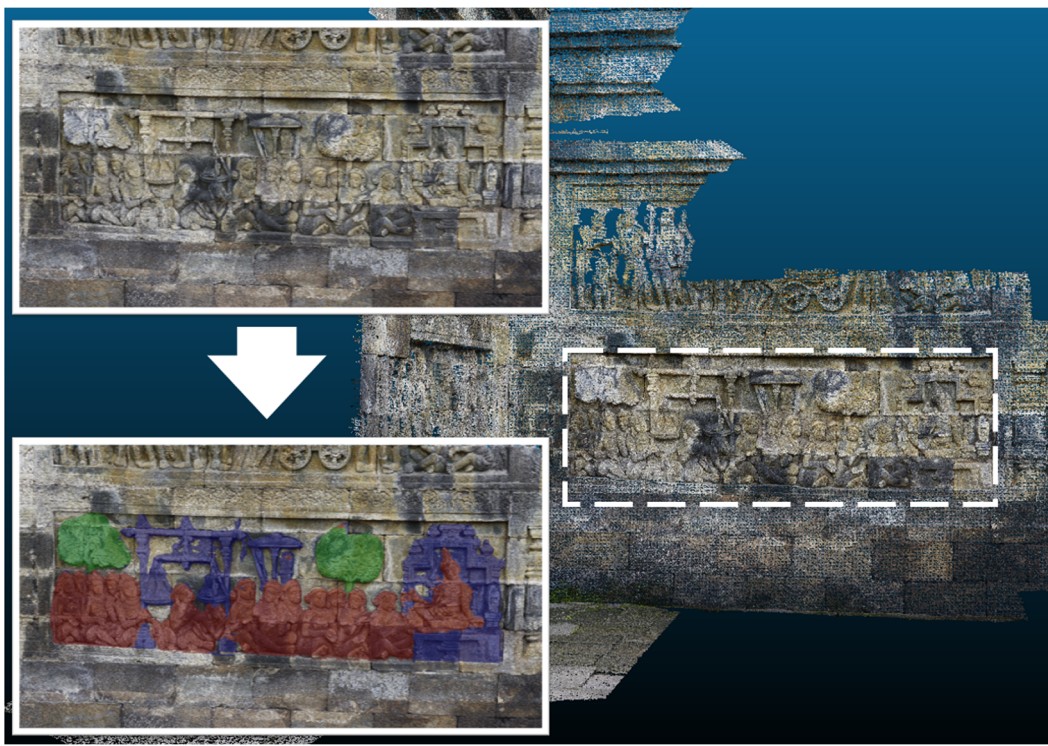

**Figure 14.** Visualization results of applying our proposed semantic segmentation framework on photogrammetric point cloud data.

## 5. Discussion

According to the prediction results in Figure 11, our results outperform those of the conventional two-dimensional semantic segmentation network. Our method is robust to relief images with mottled colors due to corrosion. The bottom 4 rows in Figure 11 show that the colors of the input RGB images are uniform, as these reliefs are not heavily corroded and the segmentation can be visually distinguished. The existing two-dimensional semantic segmentation networks are more stable for these images. However, the top 2 rows in Figure 11 show images of heavily corrupted reliefs whose colors are nonuniform. Their results in the existing two-dimensional semantic segmentation networks are strongly affected, and it is difficult to segment accurately from these unclear colors. This is proven

by the results in Table 2, where the existing networks have a low average in recall, precision, accuracy, and mIoU.

However, our results predict consistent semantic segmentation for each object in all cases and are robust to any kind of RGB image. Moreover, our semantic segmentation results make the contours much clearer. A comparison of the results of the evaluation metrics also shows that our network outperforms the other networks by more than 20 percent in all evaluations.

For the proposed method, the resolution of the output image is $3200 \times 1024$. When mapped to the point cloud, one pixel corresponds so less than 1 mm. The resolution of the segmentation results can successfully support our requirements for segmenting four categories of target objects. Expansion of segmentation categories is a plan for the continuation of this research. There are many other important categories, such as the category "ship" in Borobudur reliefs. This requires high costs of data acquisition. We plan to expand the segmentation categories to enrich semantic segmentation results in our future work.

## 6. Conclusions

In this paper, we propose a deep-learning-based, soft-edge enhanced semantic segmentation method that automatically classifies carved items (people, buildings, plants, etc.) in cultural heritage reliefs. This method is successfully applied to the bas-reliefs of the Borobudur Temple. The method is characterized by utilizing the unique feature of reliefs, which are the soft (low-curvature) edges composing boundaries of the carved items. The soft edge is extracted by our novel opacity-based edge highlighting method. We also propose a soft-edge enhanced network with a multichannel image as input, i.e., RGB images, depth maps, and soft-edge images. The soft-edge images are utilized as guide information in our soft-edge enhanced network. An intermediate prediction of the semantic edge is arranged for the soft edge information in the network. By supervised multitask learning, soft-edge information is effectively utilized for semantic segmentation. We demonstrate the accuracy of the proposed method. According to the results of the quantitative experiments, the accuracy of the semantic segmentation reaches 90%, which prominently exceeds the accuracy of other current methods. For the results of qualitative experiments, the proposed method successfully classifies people, plants, backgrounds, and other objects and maintains the clarity of the edge for segmentation and the consistency of semantic segmentation for each object. Each kind of pattern is clearly divided in the semantic results, which increases the readability of the contents of the reliefs.

In future work, we consider that, not only the semantic segmentation task, but also the object detection and instance segmentation algorithms can go further to increase the readability of relief data. In our future research, we plan to combine semantic segmentation, target detection, and instance segmentation algorithms to improve our relief visualization and analysis tool. We also hope to use our relief visualization analysis tool for digital museums, humanities research, and other applications.

**Author Contributions:** Conceptualization, S.J., J.P., L.L., K.H. and S.T.; methodology, S.J., J.P., K.H., L.L. and S.T.; software, S.J.; validation, S.J.; formal analysis, S.J.; investigation, S.J.; resources, H.Y., F.I.T., U.S. and B.; data curation, S.J., J.P. and L.L.; writing—original draft preparation, S.J.; writing—review and editing, S.J., J.P., L.L., K.H. and S.T.; visualization, S.J.; supervision, J.P., L.L. and S.T.; project administration, S.T.; funding acquisition, S.T. All authors have read and agreed to the published version of the manuscript.

**Funding:** This work is partially supported by JSPS KAKENHI Grant Number 19KK0256, 21H04903, and the Program for Asia-Japan Research Development (Ritsumeikan University, Kyoto, Japan).

**Data Availability Statement:** Not applicable.

**Acknowledgments:** In this paper, images of the Borobudur temple are presented with the permission of the Borobudur Conservation Office and Research Center for Area Studies, National Research and Innovation Agency, Indonesia. Additionally, photogrammetry data were acquired with the cooperation of the Borobudur Conservation Office, the Nara National Research Institute for Cultural Properties, the Art Research Center of Ritsumeikan University, and the Asia-Japan Research Institute of Ritsumeikan University. We deeply thank these institutions for their generous cooperation.

**Conflicts of Interest:** The authors declare no conflict of interest.

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
