# Peer review of "Semantic Segmentation for Digital Archives of Borobudur Reliefs Based on Soft-Edge Enhanced Deep Learning"

_remotesensing, doi:10.3390/rs15040956_

Round 1

Reviewer 1 Report

The main probelm I see with this paper is that the resolution of the segmentation is very low (Classes people - vegetation - buildings). It should be better explained where this low resolution is useful for the Cultural Arts or even Conservation experts. My experience tells me that these people usually require a much more detailled respresentation.

It would be sufficient to modify the Introduction and the Conclusions.

Author Response

Thank you for your insightful comments.

Reviewer 2 Report

REVISION

 The authors propose a methodology for the segmentation and visualization of elements with a high patrimonial value, acquired by a process of 3D scanning or photogrammetry. This information is important for the accurate grasp of reality. The problem is the adequate interpretation and classification of the information, reducing the processes of human manipulation. This article proposes a semantic segmentation and visualization framework that automatically classifies the elements carved in reliefs of architectural heritage, reducing barriers in the segmentation process in those cases where there is a relief where the limits of each carved element are formed by smooth edges. indistinct. It is one more step in the automation of processes, with effective results and with the goal of achieving the highest precision.

The paper scientifically exposes the methods to solve the problem, proposes an enhanced deep learning-based soft-edge network to extract the semantic labels of each relief element from multi-channel images that are projected from the 3D point clouds of the reliefs. The results are very promising since the edges of the reliefs can be clearly extracted, constituting a novel method of highlighting edges by applying an opacity process on the images. Finally, by mapping the extracted semantic labels onto 3D points of the relief elements (from the Borobudur case study), the method provides comprehensive 3D semantic segmentation results.

It would only be necessary to review the document in the marked paragraphs so that the document is better understandable, as well as correct some errors in the numbering:

Line 230: … that we proposed in our previous work. * Rev: Reference the document.

Line 234: The opacity is based on stochastic point-based rendering (SPBR) [54]. * Rev: Explain the SPBR process for transparent display of transparent images.

Line 414: We draw the semantic segmentation category labels on the 2D image and… * Rev: Explain the process of drawing the semantic segmentation category labels on the 2D image.

Line 449: The results are shown in Figure 9. * Rev: It would be number 10.

Line 479: … colors (bottom 4 rows). For images of corrupted relief (top 2 rows), their results are strongly affected. * Rev: Explain more clearly, referring to the table above.

Author Response

Thank you for your insightful comments.

Reviewer 3 Report

This paper presents a method to segment reliefs from monuments using first edge detection projection to images, image segmentation with a deep neural network and then projection of the segmentation on the 3D relief.

Overall I like the proposed approach. I find good practical applications of what they did in the 3D game industry (normal mapping on textured monuments), virtual reality, augmented reality realm. 

I suggest publication with a correction of English language.  

Author Response

Thank you for your positive and insightful comments. We made meticulous corrections to the English language in the revised manuscript.
